# Learning to Prune in Metric and Non-Metric Spaces

**Leonid Boytsov**

Carnegie Mellon University

Pittsburgh, PA, USA

srchvrs@cmu.edu

**Bilegsaikhan Naidan**

Norwegian University of Science and Technology

Trondheim, Norway

bileg@idi.ntnu.no

## Abstract

Our focus is on approximate nearest neighbor retrieval in metric and non-metric spaces. We employ a VP-tree and explore two simple yet effective learning-to-prune approaches: density estimation through sampling and "stretching" of the triangle inequality. Both methods are evaluated using data sets with metric (Euclidean) and non-metric (KL-divergence and Itakura-Saito) distance functions. Conditions on spaces where the VP-tree is applicable are discussed. The VP-tree with a learned pruner is compared against the recently proposed state-of-the-art approaches: the bbtree, the multi-probe locality sensitive hashing (LSH), and permutation methods. Our method was competitive against state-of-the-art methods and, in most cases, was more efficient for the same rank approximation quality.

## 1 Introduction

Similarity search algorithms are essential to multimedia retrieval, computational biology, and statistical machine learning. Resemblance between objects $x$ and $y$ is typically expressed in the form of a distance function $d(x, y)$, where smaller values indicate less dissimilarity. In our work we use the Euclidean distance ($L_2$), the KL-divergence ($\sum x_i \log x_i/y_i$), and the Itakura-Saito distance ($\sum x_i/y_i - \log x_i/y_i - 1$). KL-divergence is commonly used in text analysis, image classification, and machine learning [6]. Both KL-divergence and the Itakura-Saito distance belong to a class of distances called Bregman divergences.

Our interest is in the nearest neighbor (NN) search, i.e., we aim to retrieve the object $o$ that is closest to the query $q$. For the KL-divergence and other non-symmetric distances two types of NN-queries are defined. The *left* NN-query returns the object $o$ that minimizes the distance $d(o, q)$, while the *right* NN-query finds $o$ that minimizes $d(q, o)$.

The distance function can be computationally expensive. There was a considerable effort to reduce computational costs through approximating the distance function, projecting data in a low-dimensional space, and/or applying a hierarchical space decomposition. In the case of the hierarchical space decomposition, a retrieval process is a recursion that employs an "oracle" procedure. At each step of the recursion, retrieval can continue in one or more partitions. The oracle allows one to prune partitions without directly comparing the query against data points in these partitions. To this end, the oracle assesses the query and estimates which partitions may contain an answer and, therefore, should be recursively analyzed. A pruning algorithm is essentially a binary classifier. In metric spaces, one can use the classifier based on the triangle inequality. In non-metric spaces, a classifier can be learned from data.

There are numerous data structures that speedup the NN-search by creating hierarchies of partitions at index time, most notably the VP-tree [28, 31] and the KD-tree [4]. A comprehensive review of these approaches can be found in books by Zezula et al. [32] and Samet [27]. As dimensionality

increases, the filtering efficiency of space-partitioning methods decreases rapidly, which is known as the "curse of dimensionality" [30]. This happens because in high-dimensional spaces histograms of distances and 1-Lipschitz function values become concentrated [25]. The negative effect can be partially offset by creating overlapping partitions (see, e.g., [21]) and, thus, trading index size for retrieval time. The approximate NN-queries are less affected by the curse of the dimensionality, because it is possible to reduce retrieval time at the cost of missing some relevant answers [18, 9, 25]. Low-dimensional data sets embedded into a high-dimensional space do not exhibit high concentration of distances, i.e., their *intrinsic* dimensionality is low. In metric spaces, it was proposed to compute the intrinsic dimensionality as the half of the squared signal to noise ratio (for the distance distribution) [10].

A well-known approximate NN-search method is the locality sensitive hashing (LSH) [18, 17]. It is based on the idea of random projections [18, 20]. There is also an extension of the LSH for *symmetric* non-metric distances [23]. The LSH employs several hash functions: It is likely that close objects have same hash values and distant objects have different hash values. In the classic LSH index, the probability of finding an element in one hash table is small and, consequently, many hash tables are to be created during indexing. To reduce space requirements, Lv et al. proposed a multi-probe version of the LSH, which can query multiple buckets of the same hash table [22]. Performance of the LSH depends on the choice of parameters, which can be tuned to fit the distribution of data [11].

For approximate searching it was demonstrated that an early termination strategy could rely on information about distances from typical queries to their respective nearest neighbors [33, 1]. Amato et al. [1] showed that density estimates can be used to approximate a pruning function in metric spaces. They relied on a hierarchical decomposition method (an M-tree) and proposed to visit partitions in the order defined by density estimates. Chávez and Navarro [9] proposed to relax triangle-inequality based lower bounds for distances to potential nearest neighbors. The approach, which they dubbed as *stretching of the triangle inequality*, involves multiplying an exact bound by $\alpha > 1$.

Few methods were designed to work in non-metric spaces. One common indexing approach involves mapping the data to a low-dimensional Euclidean space. The goal is to find the mapping without large distortions of the original similarity measure [19, 16]. Jacobs et al. [19] review various projection methods and argue that such a coercion is often against the nature of a similarity measure, which can be, e.g., intrinsically non-symmetric. A mapping can be found using machine learning methods. This can be done either separately for each data point [12, 24] or by computing one global model [3]. There are also a number of approaches, where machine learning is used to estimate optimal parameters of classic search methods [7]. Vermorel [29] applied VP-trees to searching in undisclosed non-metric spaces without trying to learn a pruning function. Like Amato et al. [1], he proposed to visit partitions in the order defined by density estimates and employed the same early termination method as Zezula et al. [33].

Cayton [6] proposed a Bregman ball tree (bbtree), which is an exact search method for Bregman divergences. The bbtree divides data into two clusters (each covered by a Bregman ball) and recursively repeats this procedure for each cluster until the number of data points in a cluster falls below a threshold (a bucket size). At search time, the method relies on properties of Bregman divergences to compute the shortest distances to covering balls. This is an expensive iterative procedure that may require several computations of direct and inverse gradients, as well as of several distances. Additionally, Cayton [6] employed an early termination method: The algorithm can be told to stop after processing a pre-specified number of buckets. The resulting method is an approximate search procedure. Zhang et al. [34] proposed an exact search method based on estimating the maximum distance to a bounding rectangle, but it works with *left queries* only. The most efficient variant of this method relies on an optimization technique applicable only to certain decomposable Bregman divergences (a decomposable distance is a sum of values computed separately for each coordinate).

Chávez et al. [8] as well as Amato and Savino [2] independently proposed permutation-based search methods. These approximate methods do not involve learning, but, nevertheless, are applicable to non-metric spaces. At index time, $k$ pivots are selected. For every data point, we create a list, called a *permutation*, where pivots are sorted in the order of increasing distances from the data point. At query time, a rank correlation (e.g., Spearman's) is computed between the permutation of the query and permutations of data points. Candidate points, which have sufficiently small correlation values, are then compared directly with the query (by computing the original distance function). One can sequentially scan the list of permutations and compute the rank correlation between the

permutation of the query and the permutation of every data point [8]. Data points are then sorted by rank-correlation values. This approach can be improved by incremental sorting [14], storing permutations as inverted files [2], or prefix trees [13].

In this work we experiment with two approaches to learning a pruning function of the VP-tree, which to our knowledge was not attempted previously. We compare the resulting method, which can be applied to both metric and non-metric spaces, with the following state-of-the-art methods: the multi-probe LSH, permutation methods, and the bbtree.

## 2 Proposed Method

### 2.1 Classic VP-tree

In the VP-tree (also known as a ball tree) the space is partitioned with respect to a (usually randomly) chosen pivot $\pi$ [28, 31]. Assume that we have computed distances from all points to the pivot $\pi$ and $R$ is a median of these distances. The sphere centered at $\pi$ with the radius $R$ divides the space into two partitions, each of which contains approximately half of all points. Points inside the pivot-centered sphere are placed into the left subtree, while points outside the pivot-centered sphere are placed into the right subtree (points on the border may be placed arbitrarily). The search algorithm proceeds recursively. When the number of data points is below a certain threshold (the bucket size), these data points are stored as a single bucket. The obtained hierarchical partition is represented by the binary tree, where buckets are leaves.

The NN-search is a recursive traversal procedure that starts from the root of the tree and iteratively updates the distance $r$ to the closest object found. When it reaches a bucket (i.e., a leaf), bucket elements are searched sequentially. Each internal node stores the pivot $\pi$ and the radius $R$. In a metric space with the distance $d(x, y)$, we use the triangle inequality to prune the search space. We visit:

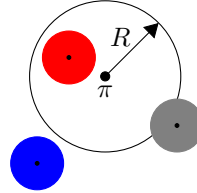

Figure 1: Three types of query balls in the VP-tree. The black circle (centered at the pivot $\pi$) is the sphere that divides the space.

- *only* the left subtree if $d(\pi, q) < R - r$;
- *only* the right subtree if $d(\pi, q) > R + r$;
- both subtrees if $R - r \leq d(\pi, q) \leq R + r$.

In the third case, we first visit the partition that contains $q$. These three cases are illustrated in Fig. 1. Let $D_{\pi,R}(x) = |R - x|$. Then we need to visit both partitions if and only if $r \geq D_{\pi,R}(d(\pi, q))$. If $r < D_{\pi,R}(d(\pi, q))$, we visit only the partition containing the query point. In this case, we prune the other partition. Pruning is a *classification task* with three classes, where the prediction function is defined through $D_{\pi,R}(x)$. The only argument of this function is a distance between the pivot and the query, i.e., $d(\pi, q)$. The function value is equal to the maximum radius of the query ball that fits inside the partition containing the query (see the red and the blue sample balls in Fig. 1).

### 2.2 Approximating $D_{\pi,R}(x)$ with a Piece-wise Linear Function

In Section 2 of the supplemental materials, we describe a straightforward sampling algorithm to learn the decision function $D_{\pi,R}(x)$ for every pivot $\pi$. This method turned out to be inferior to most state-of-the-art approaches. It is, nevertheless, instructive to examine the decision functions $D_{\pi,R}(x)$ learned by sampling for the Euclidean distance and KL-divergence (see Table 1 for details on data sets).

Each point in Fig. 2a-2c is a value of the decision function obtained by sampling. Blue curves are fit to these points. For the Euclidean data (Fig. 2a), $D_{\pi,R}(x)$ resembles a piece-wise linear function approximately equal to $|R - x|$. For the KL-divergence data (Fig. 2b and 2c), $D_{\pi,R}(x)$ looks like a U-shape and a hockey-stick curve, respectively. Yet, most data points concentrate around the median (denoted by a dashed red line). In this area, a piece-wise linear approximation of $D_{\pi,R}(x)$ could

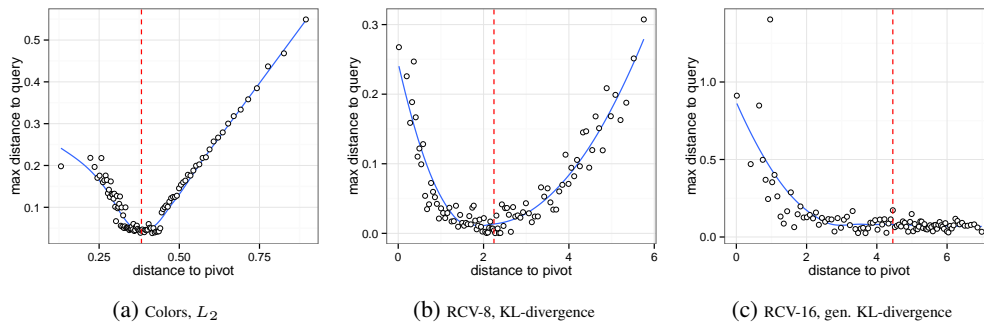

(a) Colors, $L_2$        (b) RCV-8, KL-divergence        (c) RCV-16, gen. KL-divergence

Figure 2: The empirically obtained decision function $D_{\pi,R}(x)$. Each point is a value of the function learned by sampling (see Section 2 of the supplemental materials). Blue curves are fit to these points. The red dashed line denotes a median distance $R$ from data set points to the pivot $\pi$.

still be reasonable. Formally, we define the decision function as:

$$D_{\pi,R}(x) = \begin{cases} \alpha_{left}|x - R|, & \text{if } x \leq R \\ \alpha_{right}|x - R|, & \text{if } x \geq R \end{cases} \tag{1}$$

Once we obtain the values of $\alpha_{left}$ and $\alpha_{right}$ that permit near exact searching, we can induce more aggressive pruning by increasing $\alpha_{left}$ and/or $\alpha_{right}$, thus, exploring trade-offs between retrieval efficiency and effectiveness. This is similar to stretching of the triangle inequality proposed by Chávez and Navarro [9].

Optimal $\alpha_{left}$ and $\alpha_{right}$ are determined using a grid search. To this end, we index a small subset of the data points and seek to obtain parameters that give the shortest retrieval time at a specified recall threshold. The grid search is initialized by values $a$ and $b$. Then, recall values and retrieval times for all $\alpha_{left} = a\rho^{i/m-0.5}$ and $\alpha_{right} = b\rho^{j/m-0.5}$ are obtained ($1 \leq i, j \leq m$). The values of $m$ and $\rho$ are chosen so that: (1) the grid step is reasonably small (i.e., $\rho^{1/m}$ is close to one); (2) the search space is manageable (i.e., $m$ is not large).

If the obtained recall values are considerably larger than a specified threshold, the procedure repeats the grid search using larger values of $a$ and $b$. Similarly, if the recall is not sufficient, the values of $a$ and $b$ are decreased and the grid search is repeated. One can see that the perfect recall can be achieved with $\alpha_{left} = 0$ and $\alpha_{right} = 0$. In this case, no pruning is done and the data set is searched sequentially. Values of $\alpha_{left} = \infty$ and $\alpha_{right} = \infty$ represent an (almost) zero recall, because one of the partitions is always pruned.

## 2.3 Applicability Conditions

It is possible to apply the classic VP-tree algorithm only to data sets such that $D_{\pi,R}(d(\pi,q)) > 0$ when $d(\pi,q) \neq R$. In a relaxed version of this applicability condition, we require that $D_{\pi,R}(d(\pi,q)) > 0$ for almost all queries and a large subset of data points. More formally:

**Property 1.** *For any pivot $\pi$, probability $\alpha$, and distance $x \neq R$, there exists a radius $r > 0$ such that, if two randomly selected points $q$ (a potential query) and $u$ (a potential nearest neighbor) satisfy $d(\pi,q) = x$ and $d(u,q) \leq r$, then both $p$ and $q$ belong to the same partition (defined by $\pi$ and $R$) with a probability at least $\alpha$.*

The Property 1, which is true for all metric spaces due to the triangle inequality, holds in the case of the KL-divergence and data points $u$ sampled randomly and uniformly from the simplex $\{x_i | x_i \geq 0, \sum x_i = 1\}$. The proof, which is given in Section 1 of supplemental materials, can be trivially extended to other non-negative distance functions $d(x,y) \geq 0$ (e.g., to the Itakura-Saito distance) that satisfy (additional compactness requirements may be required): (1) $d(x,y) = 0 \Leftrightarrow x = y$; (2) the set of discontinuities of $d(x,y)$ has measure zero in $L_2$. This suggests that the VP-tree could be applicable to a wide class of non-metric spaces.

Table 1: Description of the data sets

| Name | $d(x, y)$ | Data set size | Dimensionality | Source |
|------|-----------|---------------|----------------|--------|
| Colors | $L_2$ | $1.1 \cdot 10^5$ | 112 | Metric Space Library[1] |
| RCV-$i$ | KL-div, $L_2$ | $0.5 \cdot 10^6$ | $i \in \{8, 16, 32, 128, 256\}$ | Cayton [6] |
| SIFT-signat. | KL-div, $L_2$ | $1 \cdot 10^4$ | 1111 | Cayton [6] |
| Uniform | $L_2$ | $0.5 \cdot 10^6$ | 64 | Sampled from $U^{64}[0, 1]$ |

## 3 Experiments

We run experiments on a Linux server equipped with Intel Core i7 2600 (3.40 GHz, 8192 KB of L3 CPU cache) and 16 GB of DDR3 RAM (transfer rate is 20GB/sec). The software (including scripts that can be used to reproduce our results) is available online, as a part of the *Non-Metric Space Library*[2] [5]. The code was written in C++, compiled using GNU C++ 4.7 (optimization flag -Ofast), and executed in a single thread. SIMD instructions were enabled using the flags -msse2 -msse4.1 -mssse3.

All distance and rank correlation functions are highly optimized and employ SIMD instructions. Vector elements were single-precision numbers. For the KL-divergence, though, we also implemented a slower version, which computes logarithms on-line, i.e., for each distance computation. The faster version computes logarithms of vector elements off-line, i.e., during indexing, and stores with the vectors. Additionally, we need to compute logarithms of query vector elements, but this is done only once per query. The optimized implementation is about 30x times faster than the slower one.

The data sets are described in Table 1. Each data set is randomly divided into two parts. The smaller part (containing 1,000 elements) is used as queries, while the larger part is indexed. This procedure is repeated 5 times (for each data sets) and results are aggregated using a classic fixed-effect model [15]. *Improvement in efficiency* due to indexing is measured as a reduction in retrieval time compared to a sequential, i.e., exhaustive, search. The effectiveness was measured using a simple rank error metric proposed by Cayton [6]. It is equal to the number of NN-points closer to the query than the nearest point returned by the search method. This metric is appropriate mostly for 1-NN queries. We present results only for left queries, but we also verified that in the case of right queries the VP-tree provides similar effectiveness/efficiency trade-offs. We ran benchmarks for $L_2$, the KL-divergence,[3] and the Itakura-Saito distance. Implemented methods included:

- The novel search algorithm based on the VP-tree and a piece-wise linear approximation for $D_{\pi,R}(x)$ as described in Section 2.2. The parameters of the grid search algorithm were: $m = 7$ and $\rho = 8$.

- The permutation method with incremental sorting [14]. The near-optimal performance was obtained by using 16 pivots.

- The permutation prefix index, where permutation profiles are stored in a prefix tree of limited depth [13]. We used 16 pivots and the maximal prefix length 4 (again selected for best performance).

- The bbtree [6], which is designed for Bregman divergences, and, thus, it was not used with $L_2$.

- The multi-probe LSH, which is designed to work only for $L_2$. The implementation employs the LSHKit, [4] which is embedded in the *Non-Metric Space Library*. The best-performing configuration that we could find used 10 probes and 50 hash tables. The remaining parameters were selected automatically using the cost model proposed by Dong et al. [11].

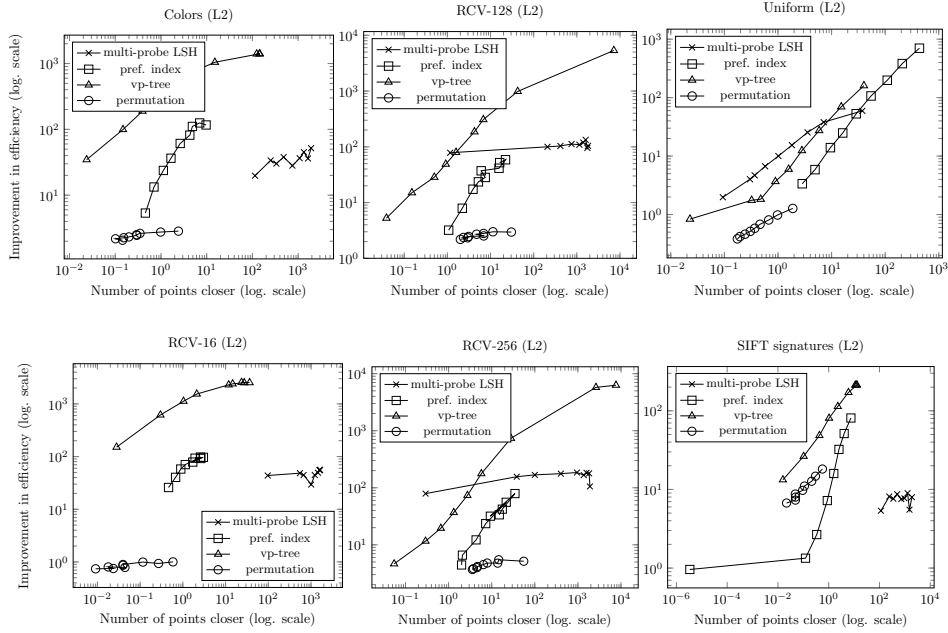

Figure 3: Performance of NN-search for $L_2$

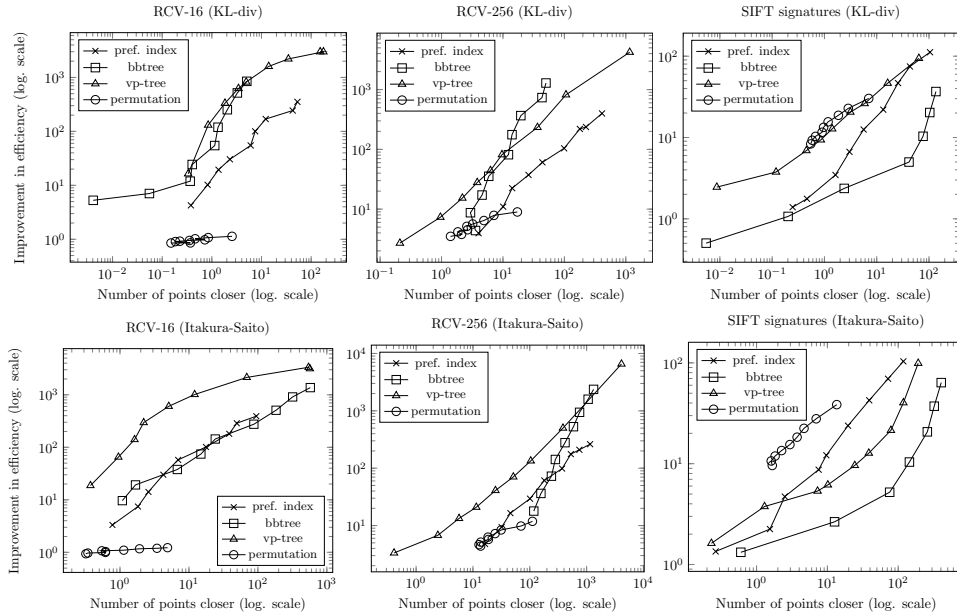

Figure 4: Performance of NN-search for the KL-divergence and Itakura-Saito distance

For the bbtree and the VP-tree, vectors in the same bucket were stored in contiguous chunks of memory (allowing for about 1.5-2x reduction in retrieval times). It is typically more efficient to search elements of a small bucket sequentially, rather than using an index. A near-optimal performance was obtained with 50 elements in a bucket. The same optimization approach was also used for both permutation methods.

Several parameters were manually selected to achieve various effectiveness/efficiency trade-offs. They included: the minimal number/percentage of candidates in permutation methods, the desired

Table 2: Improvement in efficiency and retrieval time (ms) for the bbtree without early termination

| Data set | RCV-16 | | RCV-32 | | RCV-128 | | RCV-256 | | SIFT sign. | |
|---|---|---|---|---|---|---|---|---|---|---|
| | impr. | time | impr. | time | impr. | time | impr. | time | impr. | time |
| Slow KL-divergence | 15.7 | 8 | 6.7 | 36 | 1.6 | 613 | 1.1 | 1700 | 0.9 | 164 |
| Fast KL-divergence | 4.6 | 2.5 | 1.9 | 9.6 | 0.5 | 108 | 0.4 | 274 | 0.4 | 18 |

recall in the multi-probe LSH and in the VP-tree, as well as the maximum number of processed buckets in the bbtree.

The results for $L_2$ are given in Fig. 3. Even though a representational dimensionality of the Uniform data set is only 64, it has the highest intrinsic dimensionality among all sets in Table 1 (according to the definition of Chávez et al. [10]). Thus, for the Uniform data set, no method achieved more than a 10x speedup over sequential searching without substantial quality degradation. For instance, for the VP-tree, a 160x speedup was only possible, when a retrieved object was a 40-th nearest neighbor (on average) instead of the first one. The multi-probe LSH can be twice as fast as the VP-tree at the expense of having a 4.7x larger index. All the remaining data sets have low or moderate intrinsic dimensionality (smaller than eight). For example, the SIFT signatures have the representational dimensionality of 1111, but the intrinsic dimensionality is only four. For data sets with low and moderate intrinsic dimensionality, the VP-tree is faster than the other methods most of the time. For the data sets Colors and RCV-16 there is a two orders of magnitude difference.

The results for the KL-divergence and Itakura-Saito distance are summarized in Fig. 4. The bb-tree is never substantially faster than the VP-tree, while being up to an order of magnitude slower for RCV-16 and RCV-256 in the case of Itakura-Saito distance. Similar to results in $L_2$, in most cases, the VP-tree is at least as fast as other methods. Yet, for the SIFT signatures data set and the Itakura-Saito distance, permutation methods can be twice as fast.

Additional analysis has showed that the VP-tree is a good rank-approximation method, but it is not necessarily the best approach in terms of recall. When the VP-tree misses the nearest neighbor, it often returns the second nearest or the third nearest neighbor instead. However, when other examined methods miss the nearest neighbor, they frequently return elements that are far from the true result. For example, the multi-probe LSH may return a true nearest neighbor 50% of the time, and 50% of the time it would return the 100-th nearest neighbor. This observation about the LSH is in line with previous findings [26].

Finally, we measured improvement in efficiency (over exhaustive search) for the bbtree, where the early termination algorithm was disabled. This was done using both the slow and the fast implementation of the KL-divergence. The results are given in Table 2. Improvements in efficiency for the case of the slower KL-divergence (reported in the first row) are consistent with those reported by Cayton [6]. The second row shows improvements in efficiency for the case of the faster KL-divergence and these improvements are substantially smaller than those reported in the first row, despite the fact that using the faster KL-divergence greatly reduces retrieval times. The reason is that the pruning algorithm of the bbtree is quite expensive. It involves computations of logarithms/exponents for coordinates of unknown vectors, and, thus, these computations cannot be deferred to index time.

## 4   Discussion and conclusions

We evaluated two simple yet effective learning-to-prune methods and showed that the resulting approach was competitive against state-of-the-art methods in both metric and non-metric spaces. In most cases, this method provided better trade-offs between rank approximation quality and retrieval speed. For datasets with low or moderate intrinsic dimensionality, the VP-tree could be one-two orders of magnitude faster than other methods (for the same rank approximation quality). We discussed applicability of our method (a VP-tree with the learned pruner) and proved a theorem supporting the point of view that our method can be applicable to a class of non-metric distances, which includes

the KL-divergence. We also showed that a simple trick of pre-computing logarithms at index time substantially improved performance of existing methods (e.g., bbtree) for the studied distances.

It should be possible to improve over basic learning-to-prune methods (employed in this work) using: (1) a better pivot-selection strategy [31]; (2) a more sophisticated sampling strategy; (3) a more accurate (non-linear) approximation for the decision function $D_{\pi,R}(x)$ (see section 2.1).

## 5   Acknowledgements

We thank Lawrence Cayton for providing the data sets, the bbtree code, and answering our questions; Anna Belova for checking the proof of Property 1 (supplemental materials) and editing the paper.

## Footnotes

[2] https://github.com/searchivarius/NonMetricSpaceLib

[3] In the case of SIFT signatures, we use generalized KL-divergence (similarly to Cayton).

[4] Downloaded from http://lshkit.sourceforge.net/

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
