[Supplementary Material · nips2013_suppl.pdf]

# Learning to Prune in Metric and Non-Metric Spaces (supplemental material)

**Leonid Boytsov**

Carnegie Mellon University

Pittsburgh, PA, USA

srchvrs@cmu.edu

**Bilegsaikhan Naidan**

Norwegian University of Science and Technology

Trondheim, Norway

bileg@idi.ntnu.no

## 1 Introduction

This short note supplements the paper "Learning to Prune in Metric and Non-Metric Spaces" [3]. We aim to provide a theoretical justification for applicability of our approach (the VP-tree with a learned pruner) to a class of non-metric spaces, which includes the KL-divergence and the Itakura-Saito distance. In addition, we describe a simple algorithm to learn the decision function through sampling.

## 2 Applicability Conditions

**Theorem 1.** *For any pivot $\pi$, probability $\alpha$, and distance $x \neq R$, there exists a radius $r > 0$ such that, if two randomly selected points $q$ (a potential query) and $u$ (a potential nearest neighbor) satisfy $d(\pi, q) = x$ and $d(u, q) \leq r$, then both $p$ and $q$ belong to the same partition (defined by $\pi$ and $R$) with a probability at least $\alpha$.*

Theorem 1, which is true for all metric spaces, holds in the case of KL-divergence and data points $u$ sampled randomly and uniformly from the simplex $\{x_i | x_i \geq 0, \sum x_i = 1\}$.

**Note 1.** *This theorem is trivially extended to many other* non-negative *distance functions $d(x,y)$ that satisfy:*

- *$d(x,y) \geq 0$ and $d(x,y) = 0 \Leftrightarrow x = y$;*

- *$d(x,y)$ is continuous except for a set of measure zero;*

In particular, the theorem holds for the Itakura-Saito distance. Note, though, that these conditions are not *sufficient*, because one may need to make additional compactness requirements. For the example, the proof for the KL-divergence relies on the fact that the distance is defined on the compact Euclidean subset.

*Proof.* It is easy to show that for any $\alpha$ there exists $\epsilon > 0$ such that all coordinates of the randomly selected vector are $\geq \epsilon$ with a probability at least $\alpha$. Further, we consider the compact set of vectors (it is compact with respect to $L_2$):

$$S(\epsilon) = \{y | 1 \geq y_i \geq \epsilon, \sum y_i = 1\}$$

The KL-divergence is defined as:

$$KL(x,y) = \sum_i x_i \log \frac{x_i}{y_i}$$

For any $y \in S(\epsilon)$, $y_i \geq \epsilon$. Thus, $KL(x, y)$ is a continuous function of both arguments on $S \times S$. Points outside $S$ are encountered with probability $1 - \alpha$, which can be made arbitrarily small by selecting a sufficiently small $\epsilon$.

For the sake of contradiction, we assume that, no matter how small is the query radius, there is a query ball with the center in $S$ at distance $r$ from the pivot. In addition, there are points inside query balls that belong to both partitions as well as to $S$. This can be seen as an adversarial game, where we select progressively decreasing radii $r_n \to 0$. For each $r_n$ our adversary finds the query ball with the center $q_n \in S$ and the radius $\leq r_n$ such that (1) $KL(\pi, q_n) = x$ and (2) the query ball intersects both space partitions and $S$. To demonstrate the latter, our adversary provides us with points $u_n^+$ and $u^-$ that lie inside the query ball and belong to different space partitions. Note that $q_n$, $u_n^+$, and $u^-$ should all belong to $S$.

Formally, there exists a sequence of radii $r_n \to 0$, the sequence of query ball centers $q_n$, and sequences of points $u_n^+$, $u_n^-$ such that:

$$KL(\pi, q_n) = x,$$

$$KL(u_n^+, q_n) \leq r_n \text{ and } KL(u_n^-, q_n) \leq r_n,$$

but

$$KL(\pi, u_n^-) < R \text{ and } KL(\pi, u_n^+) > R. \tag{1}$$

The sequence $(q_n, u_n^+, u_n^-)$ is defined on a Cartesian product $S \times S \times S$, which is compact due to Tychonoff's theorem. Because the Cartesian product is compact, we can assume that $(q_n, u_n^-, u_n^+)$ is a converging sequence and sequences $q_n$, $u_n^-$, $u_n^+$ converge as well: [1]:

$$(q_n, u_n^-, u_n^+) \to (q, u^-, u^+) \tag{2}$$

From Eq. 1-2 and continuity of the function $KL(x, y)$ on $S \times S$, we obtain:

$$KL(q, u^+) = KL(q, u^-) = 0, \tag{3}$$

$$KL(\pi, u^+) \geq R, \; KL(\pi, u^-) \leq R. \tag{4}$$

From properties of the KL-divergence and Eq. 2, it follows that $u^+ = q = u^-$. By applying $u^+ = q = u^-$ to Eq. 4, we get that $R \leq KL(\pi, q) \leq R$ and, thus, that:

$$KL(\pi, q) = R. \tag{5}$$

Again, from continuity of $KL(x, y)$, $KL(\pi, q_n) = x$ and $q_n \to q$, we obtain that $KL(\pi, q) = x$. Because $x \neq R$ this conclusion contradicts to Eq. 5.

We obtained a contradiction, which demonstrates that (almost) all sufficiently small query balls at distance $r \neq R$ from the pivot lie (for the most part) in either the left or the right partition. The exceptions are query balls centered outside $S$, or query ball parts that don't belong to $S$. Yet, as noted previously, it is possible to select $S$ such that a probability of drawing a point from $S$ will be arbitrarily close to 1. This observation finishes the proof of the theorem. □

## 3 Estimating Decision Function $D_{\pi,R}(x)$ Through Sampling

It is possible to estimate $D_{\pi,R}(x)$ (defined in Section 2.2 of [3]) through sampling. Note that the resulting search method would not be exact. A straightforward sampling method involves random and independent selection of points $q_i$ and $u_i$ from the data set. Two cases are possible depending on whether $q_i$ and $u_i$ belong to the same partition. Consider the case when $q_i$ and $u_i$ belong to different partitions. This represents the gray ball from Fig. 1. Thus, we learn that there may exist multiple pairs of points (different from $q_i$) within the distance $r = d(u_i, q_i)$ from $q_i$ situated in different partitions. Thus, we can be absolutely sure that $D_{\pi,R}(d(\pi, q_i)) \leq d(u_i, q_i)$.

In the case when $u_i$ is in the same partitions as $q_i$, we cannot, however, infer that $D_{\pi,R}(d(\pi, q_i)) > d(u_i, q_i)$. Indeed, there could exist a nearest-neighbor $u_j$, not encountered by the sampling procedure, belonging to a *different* space partition than $q_i$, but, nevertheless, satisfying: $d(u_j, q_j) \leq d(u_i, q_i)$. If we use $q_i$ as a query and set $D_{\pi,R}(d(\pi, q_i))$ to be larger than $d(u_i, q_i)$, the partition containing $u_j$ will be pruned and, consequently, $u_j$ will not be found.

Figure 1: Three types of query balls in the VP-tree. The black circle (centered at the pivot $\pi$) is the sphere that divides the space.

By repeating the sampling procedure multiple times and smoothing results (e.g., by fitting a curve or learning a regression model), we can obtain an estimate for the upper bound of $D_{\pi,R}(x)$. There are several problems with this approach. First, due to the concentration of measure, $d(\pi, q_i)$ is close to $R$ for most sampled points. Thus, $D_{\pi,R}(x)$ will be properly estimated only for values $x \approx R$. Second, it does not allow us to trade search effectiveness for efficiency.

The underlying principle of an improved sampling method is to divide the $xy$-plane, which represents the plot of the function $D_{\pi,R}(x)$, into cells. This improved sampling method works as follows:

- We compile the distribution of distances $d(\pi, q_i)$ (using all data points) and divide it into 50-500 quantiles. These "horizontal" quantiles represent the division of the $xy$-plane into vertical bars. Then, several pseudo-queries $q_i$ are randomly picked from each horizontal quantile.

- For each pseudo-query $q_i$, $K \approx 100$ pseudo near-neighbors are randomly selected from the data set. We also implemented an approach where $K$ *true* near-neighbors are obtained by exhaustively searching the data set (an idea proposed by Athitsos et al. [2]). This method is computationally expensive, but it did not result in substantial improvements.

- Now each vertical bar contains a number of pseudo near neighbors. We compute the bar-specific distributions of distances from $q_i$ to these points and divide each of the distributions into 100-1000 "vertical" quantiles. This step finalizes a division of the $xy$-plane into rectangular cells.

To estimate $D_{\pi,R}(x)$, we find the vertical bar containing the point $(x, 0)$. Then, we start scanning the cells belonging to this bar in the bottom-up fashion. The algorithm keeps two counters. The first counter $N_{all}$ is a total number of of pseudo near neighbors contained in the visited cells. The second counter $N_{diff}$ is the number neighbors that belong to a different partition than their respective pseudo queries (i.e., the number of situations when we have the gray pseudo query ball, see Fig. 1.). We stop when $N_{diff}$ becomes larger than $\gamma N_{all}$ for some threshold value $\gamma$. A $y$-coordinate corresponding to the last visited cell is used as an estimate for $D_{\pi,R}(x)$: One can use the minimum, the maximum, or any intermediate $y$-coordinate of points inside the last visited cell. The threshold $\gamma$ is selected empirically. In that, highest recall values (and slowest speeds) are obtained for $\gamma = 0$. Unlike previous efforts, see e.g. [1], our sampling algorithm estimates $D_{\pi,R}(x)$ for every pivot $\pi$ (rather than one global distribution) and, thus, it may better adapt to specifics of data partitions induced by pivots.

## Footnotes

[1] If a space $X$ is compact any sequence contains a converging subsequence with the limit in $X$.