[Reviews · NeurIPS 2013]

Submitted by Assigned_Reviewer_5

This paper studies how to improve the computational efficiency of approximate nearest neighbor retrieval methods. The paper itself has been well summarized in the abstract. It is well-written and interesting. Moreover, a key contribution is Property 1 where the authors show their proposed method is applicable to what kind of spaces.

However, the contribution is perhaps not closely related to learning and neural computing. I do not see learning plays an important role in the proposed method. The learning in this paper includes simple sampling and approximating piecewise linear function in very low dimensional spaces. As a consequence, this paper may not match the research interests of NIPS very well to me.

** Comment after authors' feedback **

The authors have clarified my concerns. Now I see why learning cannot be absent for this method, and why learning in this method should be simple (in fact learning for the underlying problem should be simple).
Summary: This paper studies how to improve the computational efficiency of approximate nearest neighbor retrieval methods. It is well-written and interesting.

Submitted by Assigned_Reviewer_6

Summary: The paper presents a method that learns a pruning algorithm for a VP-tree, in non-metric spaces. The idea is to estimate the decision function of the approximate nearest neighbor search in the VP-tree by sampling, and approximating it with a piecewise linear function. The learning to prune method is validated for the search efficiency against relevant baselines for prunning, and outperforms them substantially when the intrinsic dimensionality of the data is small.

Clarity: The paper is mostly clearly written but sometimes does not really go into explaining the implementation details and the choice of some parameters (for example, why choose K=100, m=7, rho=8 and the bucket size = 10^5? Line 185,227,315)

Originality: Learning to approximate the approximate nearest neighbor classification on a VP-tree, to the extent of my knowledge, is the first work that 'learns to prune'

Significance: Nearest neighbor method is a very fundamental topic in search or classification; thus this learning-to-prune method which approximates the nearest neighbor search with a non-linear function would be of some interest to a wide audience.

However, the datasets chosen for validation for the experiments seem rather simple and have low-dimensionality, which are far from realistic. (What is the result on the RCV-256, and SIFT for L2?) Also, whether the proposed method can achieve the desired the speed-up is not well justified for the metric space, which limits its application. For fast search in the metric space, there are existing methods that utilize LSH and embeddings. One relevant paper is as follows: [31] P. Jain, B. Kulis, K. Grauman, Fast Image Search for Learned Metrics, CVPR 2008.
Summary: The paper presents a novel learning-to-prune method that approximates the approximate nearest neighbor decision function in a VP-tree by a non-linear piecewise function learned with sampling, which results in large gains in speed-up compared to existing methods without learning. The approach of learning the decision function seems novel and the method seems to work well at least on the selected datasets, but the motivation of targeting a non-metric space specifically should be better justified, since it leaves out relevant baselines to be used for metric spaces.

Submitted by Assigned_Reviewer_7

This paper proposes to estimate the decision function to speed up the nearest neighbor retrieval process for VP-tree. More specifically the authors propose to do that via a sampling + regression with piecewise linear function. This strategy works for both metric (e.g. Euclidean space) and non-metric (eg., some Bregman divergence) space. The proposed method has been shown to be empirically faster than recently proposed state-of -the-art in most of the cases. Also, the paper discusses the applicability of VP-tree.

This method seems to be fairly reasonable, and can be viewed as an application of basic machine learning algorithms to search where brute-force evaluation is expensive. The paper is well organized and clearly written, and the experiments are convincing.

This paper is clearly written, and seems to be reasonably new and technically sound.
Summary: This paper proposes to estimate the decision function to speed up the nearest neighbor retrieval process for VP-tree. More specifically the authors propose to do that via a sampling + regression with piecewise linear function. This paper is clearly written, and seems to be reasonably new and technically sound.
Author Feedback

Author rebuttal: Dear reviewers,

Thank you for your comments!

First, learning is essential to our method, because there is no analytical solution for a prunner in generic non-metric spaces. The algorithm will not work if we remove the learning part.

Second, the learning method should be simple. Otherwise, indexing cost will be prohibitive and retrieval can be slow. In fact, we do demonstrate that a previously proposed search method (not based on a learning approach) for Bregman divergences can be slow due to using a computationally-expensive pruning function. Simple learning methods work well and it is a promise that more performance gains can be achieved with better learning algorithms and pivot-selection techniques.

Third, NN-classifiers require an efficient NN-searching method. NN-searching methods were published in proceedings of NIPS several times. Even in cases when these search methods did not involve learning at all. In particular, the bb-tree method due to Cayton. Our method can be useful as a NN-classifier and it does involve learning.

Also note that dimensionality is not exactly low. SIFT vector dimensionality is 1111. The uniform data set has a very high INTRINSIC dimensionality (in fact much higher than that of the SIFT and RCV-* data sets). We do improve over the bb-tree (KL-divergence) in many cases, even when dimensionality is high.

For the camera-ready version, we decided to expand the experimental section (and to shorten the description of the sampling approach that was not very competitive).